# Using routine healthcare data to evaluate the impact of the Medicines at Transitions Intervention (MaTI) on clinical outcomes of patients hospitalised with heart failure: protocol for the Improving the Safety and Continuity Of Medicines management at Transitions of care (ISCOMAT) cluster randomised controlled trial with embedded process evaluation, health economics evaluation and internal pilot

For numbered affiliations see end of article.

**Correspondence to**
Dr Lauren A Moreau;
l.a.moreau@leeds.ac.uk

Lauren A Moreau ![ORCID],[1] Ivana Holloway,[1] Beth Fylan ![ORCID],[2,3,4] Suzanne Hartley,[1] Bonnie Cundill,[1] Alison Fergusson,[1] Sarah Alderson ![ORCID],[5] David Phillip Alldred ![ORCID],[3,6] Chris Bojke,[7] Liz Breen ![ORCID],[2,3,4] Hanif Ismail,[2,4] Peter Gardner ![ORCID],[2,4] Ellen Mason,[1] Catherine Powell ![ORCID],[2,4] Jonathan Silcock ![ORCID],[2,4] Andrew Taylor,[8] Amanda Farrin,[1] Chris Gale,[9,10,11] On behalf of the ISCOMAT Programme Management Team

## ABSTRACT

**Introduction** Heart failure affects 26 million people globally, approximately 900 thousand people in the UK, and is increasing in incidence. Appropriate management of medicines for heart failure at the time of hospital discharge reduces readmissions, improves quality of life and increases survival. The Improving the Safety and Continuity Of Medicines management at Transitions (ISCOMAT) trial tests the effectiveness of the Medicines at Transition Intervention (MaTI), which aims to enhance self-care and increase community pharmacy involvement in the medicines management of heart failure patients.

**Methods and analysis** ISCOMAT is a parallel-group cluster randomised controlled trial, randomising 42 National Health Service trusts with cardiology wards in England on a 1:1 basis to implement the MaTI or treatment as usual. Around 2100 patients over the age of 18 admitted to hospital with heart failure with at least moderate left ventricular systolic dysfunction within the last 5 years, and planned discharge to the geographical area of the cluster will be recruited. The MaTI consists of training for staff, a toolkit for participants, transfer of discharge information to community pharmacies and a medicines reconciliation/review. Treatment as usual is determined by local policy and practices. The primary outcome is a composite of all-cause mortality

### Strengths and limitations of this study

► Improving the Safety and Continuity Of Medicines management at Transitions is a trial which evaluates a process to optimise medicines use for patients discharged from hospital with heart failure to their usual place of residence.

► The trial intervention was developed through experience-based co-design with patients and healthcare professionals.

► The trial will use nationwide patient-level health records for the primary endpoint.

► Due to intervention implementation requirements, patients were recruited from designated cardiology wards only.

► Clinical events cannot be centrally adjudicated as they arise from electronic health records. As such this is a pragmatic trial with results generalisable to the 'real-world' environment.

and heart failure-related hospitalisation at 12 months postregistration obtained from national electronic health records. The key secondary outcome is continued prescription of guideline-indicated therapies at 12 months

measured via patient-reported data and Hospital Episode Statistics. The trial contains a parallel mixed-methods process evaluation and an embedded health economics study.

**Ethics and dissemination** The study obtained approval from the Yorkshire and the Humber—Bradford Leeds Research Ethics Committee; REC reference 18/YH/0017. Findings will be disseminated via academic and policy conferences, peer-reviewed publications and social media. Amendments to the protocol are disseminated to all relevant parties as required.

**Trial registration number** ISRCTN66212970; Pre-results.

## BACKGROUND

Heart failure affects 26 million people globally,[1] approximately 900 thousand people in the UK, and is increasing in incidence.[2] People with heart failure frequently use multiple healthcare services,[3] so the economic burden of heart failure is substantial and is further compounded by high rates of hospitalisation and subsequent readmission.[4]

Heart failure may be managed through a combination of pharmacological treatments at titrated doses including ACE inhibitors (ACEi), beta-adrenoceptor antagonists (beta-blockers), mineralocorticoid/aldosterone receptor antagonists (MRA), angiotensin receptor neprilysin inhibitors, sodium-glucose transport protein 2 inhibitors and diuretics as well as implanted devices.[5] Medicines optimisation reduces hospitalisation, improves quality of life and increases survival rates.[6] However, this requires the coordinated input of several healthcare professionals, following hospitalisation, which may be subject to variation.[7] Medicines management at care transitions is problematic in healthcare systems internationally.[8] Estimated readmissions rate within 3 months of discharge for patients with heart failure are as high as 50%,[9] indicating a need to achieve improvements in the continuity of care at the time of transfer to the community.

The Medicines at Transitions Intervention (MaTI) was designed to improve the use of prescribed medicines when patients with heart failure are discharged from hospital. It was developed through qualitative research and co-design with patients and healthcare staff.[7 10] Intervention feasibility was tested in three healthcare areas with 31 participants in northern England in 2017.[11] Following the feasibility study, the intervention was refined before progressing to the main trial.[11] Following Medical Research Council guidance for the design and evaluation of complex interventions,[12] we report the protocol of the Improving the Safety and Continuity Of Medicines management at Transitions (ISCOMAT) trial, a cluster randomised controlled trial to evaluate the clinical and cost-effectiveness of the MaTI among heart failure patients.

## METHODS AND DESIGN
### Aim and objectives

The overall aim of ISCOMAT is to evaluate the effectiveness of the MaTI on the delivery of guideline recommended care and clinical outcomes for patients who are discharged from hospital with heart failure, when compared with treatment as usual (TAU).

The primary objective is to establish whether the MaTI reduces all-cause mortality and heart failure rehospitalisation measured over 12 months from registration.

The key secondary objective is to determine whether the intervention increases the proportion of participants being prescribed the composite of guideline-indicated therapies at 12 months post registration (ie, ACEi, angiotensin II receptor blocker (ARB) or salcubitril/valsartan, beta blocker and/or ivabradine and MRA).[13]

Other secondary objectives, measured at 12 months post registration unless stated otherwise, are to establish if the intervention:

▶ Increases time to all-cause mortality.
▶ Increases time to heart failure-related rehospitalisation.
▶ Increases time on guideline recommended (and indicated as above) cardiovascular medicines.
▶ Improves patient understanding of their medicines and satisfaction with medicines-related care at 2-week and 6-week postdischarge and 12-month postregistration.
▶ Increases number of days alive and out of hospital.
▶ Reduces hospitalisations (all cause, cardiovascular related and heart failure).
▶ Reduces mortality due to specific causes.
▶ Is cost-effective.

The internal pilot objective is to assess whether the trial recruitment meets the predefined progression criteria thresholds.

The objectives of the mixed-methods process evaluation are to determine intervention fidelity; the relationship between implementation/fidelity and outcomes; and the barriers or facilitators to the successful implementation and roll out of the intervention.[14]

### Trial design

This publication describes REC approved Study Protocol V.4.0 and dated 8 August 2018.

ISCOMAT is a parallel-group cluster randomised controlled trial that aims to recruit 2100 participants hospitalised with heart failure from 42 National Health Service (NHS) trusts in England (approximately 50 participants per site) over a 12-month recruitment period. Clusters are defined as a distinct geographical area and consist of the NHS trust with a cardiology ward(s) and/or a Coronary Care Unit (defined as a ward with nursing staff with specific cardiac nursing expertise) together with associated community pharmacies within the Clinical Commissioning Group boundaries in that geographical area (figure 1).

ISCOMAT was purposely designed to use the routine data sources available through providers such as NHS Digital, the Office for National Statistics and the National Heart Failure Audit. The use of routine health records data has enabled the trial to reduce the burden of data collection on research teams and patients and ensure a high level of accuracy with minimal data loss for the primary endpoint.

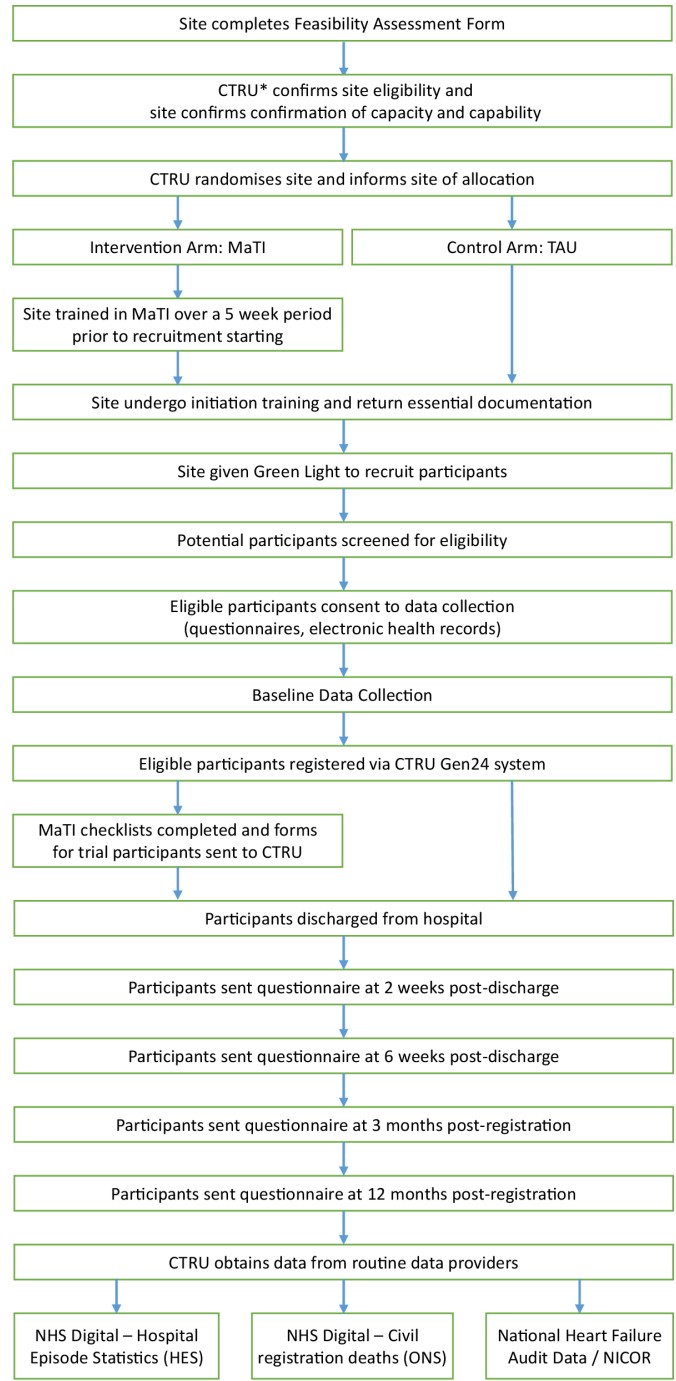

**Figure 1** Trial Process Diagram. CTRU, Clinical Trials Research Unit; NHS, National Health Service; MaTI, Medicines at Transition Intervention; TAU, treatment as usual; National Institute for Cardiovascular Outcomes Research

## Cluster eligibility

Acute NHS Trusts with accompanying cardiology wards that met all of the inclusion criteria were eligible to take part in the study as described in table 1.

## Randomisation and blinding

NHS Trusts were required to obtain all necessary local management approvals prior to randomisation and this was used to evidence consent. Randomisation was undertaken independently by the Leeds Clinical Trials Research

**Table 1** Site eligibility criteria

| Eligibility criteria | Details |
|---|---|
| Inclusion | Commitment to achieve the recruitment target. |
| | Participation in the National Heart Failure Audit.* |
| | Hospital Episode Statistics available via NHS Digital. |
| | Deemed suitable to implement and deliver the intervention (as determined by the ISCOMAT team).† |
| Exclusion | Already providing medicines management deemed to be sufficiently similar to the MaTI. |

*Participation in UK National Heart Failure Audit was required to facilitate acquiring the dataset from National Institute for Cardiovascular Outcomes Research (NICOR).
†Sites were deemed suitable where there was at least one designated cardiology ward and/or CCU and had an existing mechanism to communicate with community pharmacies for all heart failure patients (or were willing to introduce one).
CCU, Medicines at Transition Intervention; ISCOMAT, Improving the Safety and Continuity Of Medicines management at Transitions; MaTI, Medicines at Transition Intervention; NHS, National Health Service; NICOR, National Institute for Cardiovascular Outcomes Research.

Unit (CTRU). Clusters were allocated on a 1:1 basis either to implement the MaTI or TAU using minimisation with a random element, minimised by:
► Geographical region, as defined by National Institute for Health Research (NIHR) Clinical Research Networks.[15]
► Type of hospital (university hospital or non-university hospital as defined by membership of the University Hospital Association).[16]
► Method of clinical handover/information transfer to community pharmacy (electronic system, eg, PharmOutcomes[17]; or manual system, eg, standard email, fax, letter or phone).

## Participant eligibility and registration

Patient recruitment commenced approximately 4 weeks following cluster randomisation. Potential participants were screened following admission to hospital and recruited prior to discharge. Patient screening, consent, recruitment and baseline data collection was completed by research staff, who had no role in the delivery of care or treatment and, wherever possible, were blinded to the cluster allocation to mitigate selection bias.[18] Site staff were asked to self-report instances of unblinding to CTRU. This was monitored at monthly Trial Management Group Meetings (TMG) and annual Trial Steering Committee (TSC) meetings. Consented participants meeting the eligibility criteria described in table 2 are included in the trial.

Documented reasons for ineligibility or declining participation were closely monitored by the trial team as part of a regular review of recruitment progress, to ensure generalisability of study results in accordance with Consolidated Standards of Reporting Trials (CONSORT) reporting guidelines, and to highlight issues in identifying or recruiting patients during the internal pilot.

**Table 2** Participant eligibility criteria

| Eligibility criteria | Details |
|---|---|
| Inclusion | Admitted or transferred to a ward participating in the ISCOMAT trial |
| | Heart failure with evidence of at least moderate left ventricular systolic dysfunction* confirmed within the last 5 years. |
| | Aged 18 years or over at hospital admission. |
| | Planned discharged from recruiting hospital to their home (defined by usual place of residence) or a care home. |
| | Planned discharge within geographical area of that cluster. |
| | Capacity to provide informed consent. |
| | Provide informed consent. |
| Exclusion | Patients in a terminal phase of illness/end of life care pathway who were not expected to survive beyond 6 weeks postdischarge. |
| | Patients already participating in ISCOMAT (eg, readmissions). |

*Defined as: a left ventricular ejection fraction of less than or equal to 44%, quantified via echocardiogram or the equivalent of this, if quantified by a different imaging modality.
ISCOMAT, Improving the Safety and Continuity Of Medicines management at Transitions.

Following confirmation of written informed consent and eligibility, participants were registered into the trial by an authorised member of staff at the trial research site, via a central automated 24-hour online registration system hosted at CTRU.

## Trial status

The first trial site was randomised on 16 April 2018, opened to recruitment on 1 June 2018 and the first participant was recruited on 8 June 2018. A total of 44 clusters were randomised, of which 43 opened to recruitment and registered at least one participant. Overall 1641 participants consented to take part. Recruitment was due to end on 31 December 2019 but was extended to 31 December 2020. Following discussion with the sponsor and in line with government guidance, recruitment at all sites paused on 18 March 2020 due to the COVID-19 pandemic and following consultation with the TSC/Programme Steering Committee/funder the decision was taken to close ISCOMAT to recruitment on 28 July 2020.

## Intervention

The MaTI was designed to be implemented by differently specialised professional clinical teams, including nurses and pharmacists. The intervention was initiated by the discharge nurse (or equivalent) and continued postdischarge through involvement of the community pharmacist. MaTI comprises of:

1. Patient-held personalisable medicines toolkit containing: information about heart failure, medicines and their effects; details of the patients' healthcare team; a checklist of medicines-related activity that should occur in hospital before discharge, at discharge and once the patient has returned home; a traffic-light guide to symptoms plus actions to take if symptoms worsen. This will be delivered to patients face to face by a nurse

of pharmacist from the cardiology ward as part of routine care at intervention sites. See online supplemental files 1 and 2 MaTI form A and B, for example, checklists completed at site.

2. Enhanced communication between hospitals and the patients' community pharmacists through transfer of the patient's discharge letter and medicines list to community pharmacy, with a request for the community pharmacist to reconcile the patient's medicines and invite the patient for a medicines use review (MUR) or a discussion about their medicines. See online supplemental file 3 of community pharmacy cover letter.

The intervention was delivered to all eligible patients as part of routine care. Clinical staff were not aware which patients have consented to data collection and evaluation.

## Intervention training

Clusters randomised to the intervention were asked to identify key members of the clinical team involved in medicines management (eg, ward nurses, hospital pharmacists, primary care heart failure nurses) to complete intervention training online and in-person. The intervention team aimed to train around 5–7 staff members in each hospital site to enable the site coordinator and nominated staff to brief others about the MaTI. The online training was developed with the Centre for Pharmacy Postgraduate Education (www.cppe.ac.uk) to enable staff to gain an understanding of the MaTI. Face-to-face training explained the background to MaTI (importance, purpose, design, outcomes) and went through the supporting materials/guides designed to aid facilitation of delivery. Approximately 1 month was allowed in each cluster for training and embedding the MaTI into routine practice prior to study recruitment commencing. Each cluster identified one member of staff to adopt the 'site coordinator' role, responsible for ensuring the intervention was implemented according to the guidance and training provided. The number and role of hospital staff trained in each cluster was dependent on the structure of cardiology/pharmacy services and available staff. After 3–4 weeks, further follow-up visits were arranged to ensure adherence to the protocol.

Staff delivery of the intervention on an individual level was communicated to intervention team regularly to support ongoing implementation of the intervention. Additional support and training were available when required.

## Measuring adherence to the intervention

The TMG monitored adherence to intervention protocols every month by reviewing delivery of key intervention components across all intervention sites. Performance feedback was provided to site teams via monthly emails. Data on adherence was also reviewed at annual TSC meetings.

**Box 1    Definition of hospitalisation**

Hospitalisation is defined as an acute inpatient admission (regardless of length of stay). A hospitalisation will be considered heart failure-related if the primary reason/diagnosis for admission included International Classification of Diseases 10th revision codes (ICD-10) or procedural classifications related to heart failure

## Treatment as usual

TAU was determined by local Trust policies and practices. Variations in TAU were expected across NHS cardiology services in England and documented in site feasibility assessments, during recruitment and at the end of follow-up.

## Process evaluation

A parallel embedded process evaluation aims to inform the trial findings and aid potential future wider scale implementation. The process evaluation involved six purposively selected intervention sites using a mixed-methods design. Fidelity and barriers/enablers of implementation of MaTI were explored using observation, interviews of up to 20 patients and 40 healthcare professionals, staff surveys and trial-specific data collection on MaTI adherence.[14] A parallel mixed analysis is planned. Qualitative data are thematically analysed using Framework analysis and survey data is analysed using descriptive statistics. Data are synthesised, triangulated and mapped to the Consolidated Framework for Implementation Research[19 20] where appropriate. A full description of methods are published in Powell *et al.*[14]

## Outcome measures

The primary outcome is the binary composite endpoint of all-cause mortality and heart failure-related hospitalisation within 12 months postregistration (see box 1 for definition of hospitalisation). Mortality data are obtained from the Civil Registration Death data from the Office for National Statistics, and hospitalisation data from Hospital Episode Statistics (HES)[21] from NHS Digital. Routine data for the analysis of primary and secondary outcomes is supported by site reported and patient-reported data where appropriate.

The key secondary outcome is participants still being prescribed at least one of the guideline recommended therapies at 12 months in each of the following three groups:
► ACEI; ARB; salcubitril/valsartan.
► Beta blocker; ivabradine.
► MRA.

For patients with contraindications to any of the three groups, the endpoint will be derived with respect to groups that are indicated. Data are collected via patient-reported questionnaires collected at baseline, 3-month and 6-month postregistration. We also plan to obtain data from the NHS Business Service Authority on medicines dispensed in primary care, via NHS Digital.

The following secondary outcomes are collected over the 12-month postregistration via routine data sources:
► Individual components of the primary endpoint regarded as time to event endpoints. Time will be measured from registration to the first day of heart failure rehospitalisation or death.
► Length of time on guideline recommended cardiovascular medications.
► Days alive and out of hospital: defined as the number of days in the year beginning the day after registration that the patient spends alive and not in hospital.
► Time to cardiovascular-related hospitalisation and all cause hospitalisation.
► Mortality due to specific causes.
► Healthcare resource use.

The following secondary outcomes are measured at baseline, and 3-month and 12-month postregistration unless stated otherwise:
► Patient satisfaction with their medicines care and patient understanding of their medicines; measured via a patient experience survey at 2 and 6 weeks post-discharge, and 12 months postregistration with a 10-point Likert scale. For intervention participants data on the use of the MaTI over a 12-month postregistration period is also captured.
► EuroQol5-Dimension Health Questionnaire (three levels)[22]: measures health utility (quality of life) comprising five dimensions: mobility, self-care, usual activities, pain/discomfort and anxiety/depression. Dimension responses are combined and converted into a summary index (0 for dead, 1 for perfect health and negative values for states worse than death).
► Healthcare resource use: additional to routine datasets, we capture patient reported healthcare resource use using standard self-reported resource use questionnaires, to include hospital attendances, GP, nurse, specialist heart failure nurse and pharmacy contact.

## Data collection

The outcome assessment schedule is outlined in table 3. All personal information is processed in accordance with the Data Protection Act 2018[23] (and any successor legislation). Appropriate storage, restricted access and disposal arrangements for personal and clinical details are in place. Archiving is for a minimum of 10 years.

### Routine data

ISCOMAT will obtain data from routine providers NHS Digital (including HES and ONS data) and NICOR and the UK National Heart Failure Audit for the primary and secondary outcomes as outlined above. The trial was designed to use data from national routine sources to maximise efficiency in answering the research questions, ensure consistency across multiple sites and minimise burden on patients and research staff at hospitals.

**Table 3** Schedule of enrolment, interventions and assessments for participants

| Assessment | Type | Method of completion | Screening | Baseline | 2 weeks postdischarge | 6 weeks postdischarge | 3 months postregistration | 12 months postregistration |
|---|---|---|---|---|---|---|---|---|
| **Participant data** | | | | | | | | |
| Screening (demographics/ assessment of eligibility) | CRF | CRN/local research staff | X | | | | | |
| Consent | Consent form | Self-completion | | X | | | | |
| Eligibility | CRF | CRN/local research staff | | X | | | | |
| Demographics | CRF | CRN/local research staff | | X | | | | |
| Contact details | CRF | CRN/local research staff | | X | | | | |
| Admission/ discharge details | CRF | CRN/local research staff | | X | | | | |
| Health Related Quality of Life (EQ-5D-3L) | Questionnaire booklet (post) | Self-completion | | X | | | X | X |
| Healthcare Resource Use (in last 3 months) | Questionnaire booklet (post) | Self-completion | | X | | | X | X |
| Current heart failure medications | Questionnaire booklet (post) | Self-completion | | | | | X | X |
| Patient Experience Survey | Questionnaire booklet (post) | Self-completion | | | X | X | X | X |
| Primary outcome (all-cause mortality and Heart Failure (HF) rehospitalisation) | Routine data/ CRF | Data Access Request: Routine data providers >CTRU | | Data extracts to be agreed with data providers to allow 12 month data collection | | | | |
| Secondary outcomes | Routine data/ CRF | Data Access Request: Routine data providers >CTRU | | Data extracts to be agreed with data providers to allow 12 month data collection | | | | |
| Patient status | CRF | CTRU | | Ongoing | | | | |
| Site data | | | | | | | | |

Continued

**Table 3** Continued

| Assessment | Type | Method of completion | Timepoint | | | | | |
|---|---|---|---|---|---|---|---|---|
| | | | Screening | Baseline | 2 weeks postdischarge | 6 weeks postdischarge | 3 months postregistration | 12 months postregistration |
| Assessment | Type | Method of Completion | Baseline | | 6 months postrandomisation | | 12 months postimplementation of intervention | |
| Site questionnaire | Questionnaire Booklet | CTRU | X | | X | | X | |
| Intervention data | | | | | | | | |
| Assessment | Type | Method of completion | | 3 months postregistration | | | 6 months postimplementation of intervention | |
| Adherence/fidelity data (collection of patient checklists to confirm what happened at discharge, MUR review confirmation from the pharmacy) | CRF | CTRU | Ongoing | | | | | |
| Interviews—patients | Process evaluation | Research fellow | X | | | | | |
| Interviews—staff | Process evaluation | Research fellow | | | | | X | |
| Survey—staff | Process evaluation | Research fellow | | | | | X | |
| Observations—ward | Process evaluation | Research fellow | | | | | X | |

CRF, case report form; CRN, Clinical Research Network; CTRU, Clinical Trials Research Unit; EQ-5D-3L, EuroQol5-Dimension-3 Levels; MUR, medicines use review.

## Clinical data

Recruiting teams collected information on study eligibility and details of the participants' admission and discharge, including heart failure history, clinical details, primary aetiology for heart failure, detailed medicines information and related unexpected serious adverse events. All-cause mortality and readmission data were collected throughout the study up until 12 months post registration from hospital sites and electronic health records (NHS Digital). Consent was sought to obtain data from electronic health records—including secondary care records and NHS Digital.

## Participant-reported data

Patient self-reported baseline measures were collected in hospital inpatient setting. Patient-reported outcomes at 2-week and 6-week postdischarge, and 3-month and 12-month postregistration were collected via post. When questionnaires were not returned, the CTRU sent postal reminders.

## Intervention fidelity data and quantitative and qualitative data from observations, participant and staff interviews and surveys

Clinical teams, randomised to the intervention, returned a checklist of seven steps for each trial participant, confirming completion of intervention activity and local pharmacy information. The CTRU sought data from the participants' community pharmacy following discharge. Details about the heart failure service models and the clinical role of those staff responsible for the delivery of the MaTI were documented.

## Statistical methods
### Sample size

Sample size calculations were based on previous research[24 25] indicating that the combined primary event rate is likely to be at least 20% in control sites. With 42 clusters, an estimated intracluster correlation coefficient (ICC) of 0.01, a coefficient of variation in cluster size of 0.23, and 15% lost to follow-up (based on studies in this patient group),[26] 50 patients with heart failure in each cluster (2100 in total) are required to provide at least 80% power at 5% significance level to detect a minimal clinically important difference (MCID) of 6%.[26] The MaTI is delivered to each heart failure participant by dispersed teams across the care transition (secondary care, primary care, community pharmacy), therefore, the anticipated ICC is likely to be relatively low.

This sample size also provides power in excess of 90% to detect a MCID of 12% in the key secondary outcome even with the most conservative assumption for the control rate.

## Internal pilot

Descriptive statistics were used to evaluate progression criteria assessing site-level recruitment rates. This analysis informed study continuation beyond the internal pilot phase. The progression criterion was assessed at 6 months after recruitment commenced, based on the following

traffic light system of green (go), amber (review) and red (stop):

Green: ≥4 patients/month/NHS trust.
Amber: ≥2 but <4 patients/month/NHS trust.
Red: <2 patients/month/NHS trust.

## Data analysis

A full statistical analysis plan (SAP) will be finalised prior to any analyses. The CTRU will conduct the statistical analysis and data summaries on an intention-to-treat population, which is defined as all clusters randomised and all participants registered regardless of non-compliance with the protocol or withdrawal from the study. The study is conducted and will be reported according to the CONSORT extensions for cluster trials[27] and CONSORT-ROUTINE[28] for trials conducted using cohorts and routinely collected data. Baseline data will be summarised by treatment group and at the participant level and/or the cluster level using descriptive statistics as appropriate.

Additionally, the SAP will describe the derivation of outcomes and detail the analyses for each outcome measure. The analyses models will reflect clustering and include fixed effects for the minimisation factors with clusters as random effects (see table 4).

A complier average causal effect analysis approach to estimate the treatment effect for the primary outcome among compliant clusters will be considered depending on levels of adherence to the intervention as intended. Definitions of adherence and predefined thresholds will be outlined in the SAP and made without reference to the effectiveness data.

No formal interim analyses were planned. Blinded interim reports were presented to the TSC containing descriptive information. This included data on recruitment, follow-up, adherence, safety and data quality. These were presented by randomised arm if specifically requested as such, with treatments masked. As participants were recruited following cluster randomisation, monitoring of baseline characteristics by arm are presented to assess postrandomisation selection bias. A single final analysis is planned when all follow-up data has been entered onto trial database, and the database is cleaned and locked.

In the cost-effectiveness analysis, the primary outcome is quality-adjusted life-years compared with costs from an NHS and PSS perspective. A secondary cost-effectiveness analysis will look at the incremental cost per all-cause deaths prevented and re-hospitalisation prevented. Resource use associated with 'TAU' and the intervention will be collected through the health resource use questionnaire administered to patients. Prescription and hospitalisation data will be collected from the data linkage outputs and supported where necessary by the patient health resource use questionnaire at 3 and 12 months. Unit costs for health service resources will be obtained from national sources (eg, Personal Social Services Research Unit British National Formulary for medicines). The intervention cost will include any enhancement of

**Table 4** Data analysis plan for primary and secondary outcomes and health economic analysis

| Analysis activities | Assessed at | Statistical methods |
|---|---|---|
| Primary endpoint—all-cause mortality and heart failure-related rehospitalisation within 12 months postregistration | 12 months postregistration | The primary endpoint will be analysed using a two-level logistic regression model with participants nested within clusters, with clusters treated as random effects. The model will be adjusted for the following fixed effects: the cluster level stratification variables (level 2): geographical region, type of hospital and method of clinical handover/information transfer to community pharmacy, and patient level covariates (level 1) and other relevant known predictors of outcome. Results will be expressed as point estimates, with corresponding 95% CIs, p values. An estimate of the ICC and corresponding CI will also be presented. |
| Key secondary endpoint | 12 months postregistration | This endpoint will be analysed using a similar modelling strategy as the primary endpoint. |
| Other secondary endpoints | 12 months postregistration | Time-to-event analyses will be by multilevel shared frailty models with the event right-censored at either 12 months follow-up, date of death or date of withdrawal, whichever is earliest.<br>Days alive and out of hospital will be analysed using mixed effects Poisson regression model.[31] With data censored at either 12 months postregistration, death or withdrawal, whichever is earliest.<br>Length of time on guideline recommended cardiovascular medications and cause specific death endpoints will be summarised descriptively. |
| | 2 and 6 weeks postdischarge; 12 months postregistration | Patient understanding of their medicines and satisfaction with medicines-related care will be analysed using Ordinal logistic regression with clustering model to compare participant responses between treatment arms. |
| Cost-effectiveness analysis | 12 months postregistration | The primary outcome will be quality-adjusted life-years (QALYs). A secondary cost effectiveness analysis will look at the incremental cost per all-cause deaths prevented and rehospitalisation prevented.<br>In order to assess the long-term cost-effectiveness of the intervention, compared with treatment as usual, a decision analytical cost-effectiveness model will be used to estimate the expected incremental cost per QALY. The model will include a Value of Information Analysis to assess the value of undertaking further research to reduce decision uncertainty in the model. The perspective will be the same as the within-trial analysis but the time horizon will be the lifetime of the individual to capture the full impact of any mortality differences on the long-term cost-effectiveness. Estimation of health-related quality of life will use QALYs and discounting will be at the same rate as for the within trial analysis. Parameter uncertainty will be addressed using probabilistic sensitivity analysis and running Monte Carlo simulations. Results will meet international CHEERS standards of reporting and results will be presented as Incremental Cost-Effectiveness rations, Incremental Net Health and Monetary Benefits and Cost Effectiveness Acceptability Curves. |

ICC, intracluster correlation coefficient.

existing medicines management components and any 'new' components of the MaTI (eg, the staff training module or necessary enhancements to existing community pharmacy medicines services). We will assess uncertainty using a within trial probabilistic sensitivity analysis undertaken using Monte Carlo simulation with results presented as incremental cost-effectiveness rations and cost-effectiveness acceptability curves.

For further detail on outcome measures, see table 4.

### Data monitoring
Data are monitored for quality and completeness by the CTRU using established verification, validation and checking processes. The TMG comprising the chief investigator, CTRU team, other coapplicants and PPI representative will be responsible for the ongoing management of the trial. The TMG is responsible for any audits for trial conduct required. The TMG reports to the TSC annually. The TSC is formed with an independent chair, health economist and statisticians.

### Methods to handle missing data
Missing data are expected and the proportion of missing data will be compared between intervention and control groups. Sensitivity analyses of the primary endpoint will be conducted to assess the impact of missing data, the choice of imputation model and assumptions around the missingness mechanism, as appropriate.

### ETHICS AND DISSEMINATION
The trial received favourable REC opinion from Yorkshire and the Humber—Bradford Leeds Health Research Ethics Committee on 30 January 2018 with reference number 18/YH/0017

The trial is sponsored by Bradford Teaching Hospitals NHS Foundation Trust, coordinated by the CTRU at the University of Leeds and supported by a TMG and Programme Management Group (PMG) TSC. The TSC includes independent external experts who provide overall supervision for a trial on behalf of the Trial Sponsor and Trial Funder.

This longstanding and ongoing engagement with stakeholders provides a direct pathway to impact for the outputs of this research. Our Patient-Led Steering Group (PLSG) will inform our dissemination strategy and its members will play an active role in the format and content of academic papers

(specifically patient implications) and will present at local, regional and national conferences and meetings.

### Patient and public involvement

ISCOMAT has a Patient-Led Steering Group that has contributed throughout the programme. Patient representatives have membership of the TMG, PMG and TSC. The PLSG has played an integral role in several aspects of the study, including intervention development, the development of data collection tools and the analysis of qualitative interview data. Members of the group have also served as co-authors in programme and trial publications.

### COVID-19 pandemic and considerations for trial closure

Sites had been brought on gradually in waves across a 2-year site setup period. Sites also closed at various points throughout this period due to reaching or exceeding site-specific targets sample sizes or due to changes in capacity to deliver the intervention. Prior to the onset of the COVID-19 pandemic, we had needed to extend recruitment for a minimum of 1 year in order to try and achieve target sample size. A full report on the recruitment rate and generalisability of results will be available in the trial results paper that will be published on completion of trial analysis.

The COVID-19 adversely affected routine NHS services, and there was a decline in admissions with cardiovascular disease.[29 30] Following national NIHR guidance, sponsor decision and TSC review, the ISCOMAT trial was paused in March 2020 and formally closed to recruitment in July 2020. Uncertainty about the post-COVID landscape raised concerns for intervention relevance and deliverability, future research capacity and participant willingness to be involved in research, number of admissions and organisation of heart failure services in sites, plus external policy changes, such as MURs. Additional concerns included the effect of site withdrawals following a restart on the trial design and increased post-COVID death rates on our findings. The decision to close the trial was supported by statistical review that concluded that closing recruitment would not have a significant impact on the statistical power or analysis of the primary and key secondary outcomes, given on the proportion of the target sample size recruited to date. Furthermore, this design maximising routine data sources helped to minimise the impact of COVID-19 on trial analysis plans.

### Author affiliations

[1]Leeds Institute for Clinical Trials Research, Clinical Trials Research Unit, University of Leeds, Leeds, UK
[2]School of Pharmacy and Medical Sciences, Faculty of Life Sciences, University of Bradford, Bradford, UK
[3]NIHR Yorkshire and Humber Patient Safety Translational Research Centre, Bradford Institute for Health Research, Bradford, UK
[4]Wolfson Centre for Applied Health Research, University of Bradford, Bradford, UK
[5]Academic Unit of Primary Care, University of Leeds, Leeds, UK
[6]School of Healthcare, University of Leeds, Leeds, UK
[7]Academic Unit of Health Economics, University of Leeds, Leeds, UK
[8]ISCOMAT Patient Led Steering Group, Manchester, UK
[9]University of Leeds Leeds Institute of Cardiovascular and Metabolic Medicine, Leeds, UK
[10]Department of Cardiology, Leeds Teaching Hospitals NHS Trust, Leeds, UK
[11]Leeds Institute for Data Analytics, University of Leeds, Leeds, UK

**Acknowledgements** We would like to thank Alison Blenkinsopp, Gerry Armitage, Theo Rayner, Ian Kellar, Jan Speechley, the ISCOMAT Patient-led Steering Group, the ISCOMAT Trial Management Group, Trial Steering Committee and Programme Steering Committee.

**Contributors** IH, BF, SH, BC, AF, SA, DPA, CB, LB, HI, EM, PG, CP, JS, AT, AF and CG developed the methodology and detail of the ISCOMAT trial protocol. LAM drafted the manuscript, with all authors reviewing the draft critically for intellectual content. All authors approved the final version submitted for publication.

**Funding** The study was funded as part of a National Institute for Health Research Programme Grant for Applied Research (RP-PG-0514-20009).

**Disclaimer** The views expressed are those of the authors and not necessarily those of the NIHR. The views expressed in this paper are those of the authors and not necessarily those of the NHS, the NIHR or the Department of Health.

**Competing interests** None declared.

**Patient and public involvement** Patients and/or the public were involved in the design, or conduct, or reporting, or dissemination plans of this research. Refer to the Methods section for further details.

**Patient consent for publication** Not applicable.

**Provenance and peer review** Not commissioned; externally peer reviewed.

**Author note** The trial is sponsored by Bradford Teaching Hospitals NHS Foundation Trust.

### ORCID iDs
Lauren A Moreau http://orcid.org/0000-0002-0280-6345
Beth Fylan http://orcid.org/0000-0003-0599-4537
Sarah Alderson http://orcid.org/0000-0002-5418-0495
David Phillip Alldred http://orcid.org/0000-0002-2525-4854
Liz Breen http://orcid.org/0000-0001-5204-1187
Peter Gardner http://orcid.org/0000-0002-8799-0443
Catherine Powell http://orcid.org/0000-0001-7590-0247
Jonathan Silcock http://orcid.org/0000-0002-4920-9249

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
