## [Reviewer comments · BMJ Open]

ARTICLE DETAILS

TITLE (PROVISIONAL)	Using Routine Healthcare Data to evaluate the impact of the Medicines at Transitions Intervention (MaTI) on clinical outcomes of patients hospitalised with heart failure: protocol for the Improving the Safety and Continuity Of Medicines management at Transitions of care (ISCOMAT) cluster randomised controlled trial with embedded process evaluation, health economics evaluation and internal pilot
AUTHORS	Moreau, Lauren; Holloway, Ivana; Fylan, Beth; Hartley, Suzanne; Cundill, Bonnie; Fergusson, Alison; Alderson, Sarah; Alldred, David; Bojke, Chris; Breen, Liz; Ismail, Hanif; Gardner, Peter; Mason, Ellen; Powell, Catherine; Silcock, Jonathan; Taylor, Andrew; Farrin, Amanda; Gale, Chris

VERSION 1 – REVIEW

REVIEWER	Roubille, François Université de Montpellier
REVIEW RETURNED	16-Aug-2021

GENERAL COMMENTS	The paper entitled « Using Routine Healthcare Data to evaluate the impact of the Medicines at Transitions Intervention (MaTI) on clinical outcomes of patients hospitalised with heart failure: protocol for the Improving the Safety and Continuity Of Medicines management at Transitions of care (ISCOMAT) cluster randomised controlled trial with embedded process evaluation, health economics evaluation and internal pilot ». This is a design paper. The paper is well presented and easy to follow. The methods are extensively described. The trial addresses an important question. Main concerns • As indicated on the website, the recruitment has been completed several months before. ISRCTN - ISRCTN66212970: Improving the safety and continuity of medicines management at care transitions. It seems that the number of patients is not achieved. Page 13, the authors explain the impact of COVID, which is logical. However, among 43 sites, during 2 years, that means only 1.5 patient per month and per center. That seems to low to ensure representativity and raises concerns on the trial. What about translation? Please elaborate on this point. Similarly what about abroad in other health systems? • The authors mention COVID and explain why the design could help to minimize the impact of COVID? Although true, this should
---

	be kept only in the discussion section, as the trials was designed and had started before the pandemics.  • The intervention is well described. However, examples of documents provided to patients would be useful for the reader. Similarly, the way to provide information and check adherence of pharmacists to the program are mandatory. Minor concerns  • Pharmacists and nurses are not equivalent. Details on local organizations for discharge would be valuable.
--	---

VERSION 1 – AUTHOR RESPONSE

Reviewer comment

Response to reviewer.

As indicated on the website, the recruitment has been completed several months before. ISRCTN - ISRCTN66212970: Improving the safety and continuity of medicines management at care transitions. It seems that the number of patients is not achieved. Page 13, the authors explain the impact of COVID, which is logical. However, among 43 sites, during 2 years, that means only 1.5 patient per month and per center. That seems low to ensure representativity and raises concerns on the trial. What about translation? Please elaborate on this point.

We acknowledge that the target sample size was not achieved. However, the decision not to restart the trial following a pause in recruitment due to COVID was based on a number of different factors, the statistical power considerations being just one of these. Other considerations included both the conduct and design of the trial, which would have been impacted by the pressures and changes within the NHS caused by the pandemic. All these factors were discussed in detail with the Trial Steering Committee and the decision was endorsed by the funder on the basis that sufficient data had already been gathered to analyse the outcomes.

Regarding rate of recruitment, not all sites were open at the start of the recruitment period but were opened in waves across the two years during which the trial was open to recruitment. Sites also closed at different times throughout the recruitment period, due to reaching or exceeding site-specific target sample sizes, or changes in local capacity to deliver the intervention. Hence, the sites were open to recruitment for differing durations and therefore the calculation of the recruitment rate is not as straightforward as dividing the total sample size by number of sites and months of recruitment. Consideration about representativeness also needs to take into account the numbers presenting to the hospital wards from which patients were recruited, as well as the numbers eligible, and any contextual reasons for changes observed in recruitment. We will report fully on the recruitment rate and generalisability of the results in the trial results paper that will be submitted for publication once analysis has been completed. It is also worth noting, as outlined on page 17, that the trial did have an internal pilot that measured the ability to recruit sufficient numbers of both patients and clusters against pre-defined progression criteria 6-months after recruitment commenced, and all of these criteria were met. The Trial Management Group, Trial

Steering Committee and NIHR continued to monitor the site-level, and overall, recruitment rate throughout the trial against these progression criteria, and the recruitment projections were based on when sites were due to be randomised.

We have clarified these issues in the main text of the paper.

Similarly what about abroad in other health systems?

The authors mention COVID and explain why the design could help to minimize the impact of COVID? Although true, this should be kept only in the discussion section, as the trials was designed and had started before the pandemics.

The intervention is well described.

However, examples of documents provided to patients would be useful for the reader. Similarly, the way to provide information and check adherence of pharmacists to the program are mandatory.

Pharmacists and nurses are not equivalent. Details on local organizations for discharge would be valuable.

Thank you for this comment. We have added a sentence to the Background sentence to acknowledge this point.

Thank you for this comment. We have removed the comment from the design section to the discussion section on COVID-19 and considerations for trial closure

The details of the co-designed patient toolkit will be available in a separate forthcoming paper. We have included the intervention checklist and community pharmacy information data collection forms and the letter sent from hospital to community pharmacy for reference as supplementary files.

Thank you for this comment. We have clarified this point through adding a sentence into the Intervention section (p.11)